# Constipation might be associated with risk of allergic rhinitis: A nationwide population-based cohort study

Meng-Che Wu[1,2], Ming-Shiou Jan[3,4,5], Jeng-Yuan Chiou[6], Yu-Hsun Wang[7], James Cheng-Chung Wei[1,5,8]*

1 Institute of Medicine, Chung Shan Medical University, Taichung, Taiwan, 2 Division of Gastroenterology, Children's Medical Center, Taichung Veterans General Hospital, Taichung, Taiwan, 3 Institute of Biochemistry, Microbiology and Immunology, Chung Shan Medical University, Taichung, Taiwan, 4 Immunology Research Center, Chung Shan Medical University, Taichung, Taiwan, 5 Division of Allergy, Immunology and Rheumatology, Chung Shan Medical University Hospital, Taichung, Taiwan, 6 School of Health Policy and Management, Chung Shan Medical University, Taichung, Taiwan, 7 Department of Medical Research, Chung Shan Medical University Hospital, Taichung, Taiwan, 8 Graduate Institute of Integrated Medicine, China Medical University, Taichung, Taiwan

* jccwei@gmail.com

## Abstract

### Background

Allergic rhinitis (AR) is a burdensome respiratory disorder whose etiology and pathophysiology remain controversial and most likely multifactorial. Accumulated evidence indicates that gut dysbiosis contributes to AR via the gut-airway axis. Constipation could result in alteration of the intestinal microflora. The clinical impact of constipation on AR has not been studied. We aimed to evaluate the risk of AR in constipated patients using a nationwide longitudinal population-based cohort.

### Methods

We identified 57786 patients with constipation and 57786 matched controls between 1999 and 2013 from the Longitudinal Health Insurance Database, which is a subset of Taiwanese National Health Insurance Research Database. Propensity score analysis was used for matching age, sex, comorbidities, and medications at a ratio of 1:1. Multiple Cox regression and subgroup analyses were used to estimate the adjusted hazard ratio of AR.

### Results

The incidence of AR was 32.2 per 1,000 person-years in constipated patients, which was twice that of non-constipated patients. After adjustment for patients' age, gender, comorbidities, and medications, patients with constipation had a 2.3-fold risk of AR compared to those without constipation (adjusted hazard ratio [aHR]: 2.30; 95% CI, 2.23–2.37). In subgroup analyses, patients aged 20–39 years had a 2.24-fold higher risk of AR in the constipation cohort (aHR; 95% CI, 2.12–2.36). Patients aged <20, 40–64, and ≥65 years had a 2.09, 2.05, and 2.07-fold risk of AR in the constipation cohort, respectively (aHR; 95% CI, 1.98–

**Data Availability Statement:** Data were retrieved from the Longitudinal Health Insurance Database (LHID). The LHID is a subset of the National Health Insurance Research Database (NHIRD), a database

of all medical claims in Taiwan's NHI system. The use of NHIRD is limited to research purposes only. Only Taiwanese citizens who fulfill the requirements for conducting research projects are eligible to apply for access to the National Health Insurance Research Database (NHIRD). The dataset used in this study is held by the Taiwan Ministry of Health and Welfare (MOHW). Any researcher interested in accessing this dataset can submit an application form to the Ministry of Health and Welfare requesting access (contact via http://dep.mohw.gov.tw/DOS/np-2497-113.html).

**Funding:** The author(s) received no specific funding for this work.

**Competing interests:** The authors have declared that no competing interests exist.

2.20, 1.94–2.18, and 1.92–2.23). Also, patients with constipation had a higher likelihood of AR, regardless of sex, and with or without comorbidities including hyperlipidemia, hypertension, chronic kidney disease, chronic liver disease, diabetes, chronic obstructive pulmonary disease, rheumatoid arthritis, dyspepsia, irritable bowel syndrome, and anxiety.

## Conclusion

Constipation might be associated with an increased risk of incidental AR. It seems that physicians should keep a higher index of suspicion for AR in people with constipation. The patency issue of gut could not be ignored in patients with AR.

## Introduction

Allergic rhinitis (AR) is characterized by paroxysmal sneezing, nasal stuffiness, postnasal drainage, rhinorrhea, and itchy nose. During the past few decades, the prevalence of AR has increased dramatically around the globe [1]. It affects approximately 10 to 30% of adults and children in industrialized countries [2, 3], and is associated with economic loss and significant morbidity. AR is a burdensome respiratory disorder, but its etiology and pathophysiology have not yet been fully elucidated. Constipation is one of the most common, multifactorial gastrointestinal disorders and its median prevalence worldwide ranges from 8.2% to 32.9% [4]. Complications of constipation include anal fissures, urine or fecal incontinence, hemorrhoids, and rectal prolapse, which often increase the frequency of outpatient visits or hospitalizations, thereby increasing the cost of health insurance. It is also becoming increasingly frequent and is considered a major health issue that has a significant negative impact on the quality of life (QoL) [5]. The deleterious effect of constipation on QoL has been shown to be comparable or even more severe than other chronic conditions like inflammatory bowel disease, diabetes, rheumatoid arthritis, and hemodialysis [5].

A shred of studies have suggested that atopic disease might be linked to constipation [6–8]. Studies have reported an indirect indication of a concurrence of constipation and atopy by demonstrating a high prevalence of coexistent allergic manifestations in constipated children investigated for cow milk allergy [9, 10]. An epidemiologic survey by Tokunaga et al. showed that constipation was a relevant factor for AR development (adjusted odds ratio of 1.17) among 21802 high school students [11]. Jones et al. stated that the overlap of atopy in functional gastrointestinal disorders patients, and the risk of rhinitis in 342 constipated patients was 1.66 times higher than controls [8]. Previous research also demonstrated that prolonged stool stasis in the colon had a significant impact on the intestinal ecosystem, which could affect a variety of bowel functions, including motility and mucosal immunity [12–16]. Constipation is currently considered to be a causative factor in intestinal dysbiosis [13, 17]. For instances, a study by Khalif et al. in Ireland reported a decreased abundance in Bifidobacteria and Lactobacillus [17]. A clinical trial by Kim et al. in Korea showed a decreased abundance of Bacteroides and Bifidobacterium species, when compared to the non-constipated controls [18]. Recent research using 16S rRNA gene-based microbiome analysis documented dysbiosis of gut microbiota in constipated patients [13, 14, 19, 20]. Hence, in addition to laxatives, manipulation of the gut microbiome has increasingly been seen as a novel target for therapeutic possibilities for constipation [2, 15, 20, 21]. Likewise, differences in the composition of gut microbiome have been demonstrated when comparing AR subjects and healthy controls [22–24]. Moreover, accumulated evidence shows that intestinal dysbiosis is associated with an

increased risk of AR via gut-airway axis [2, 20, 25–27]. However, little is known about whether constipation could influence AR. Data obtained from a real-world large national longitudinal database have never been utilized for investigating this relationship. We thus hypothesized that constipation could impact the risk of AR and evaluated this hypothesis by analyzing a nationwide population-based retrospective cohort from the Taiwanese National Health Insurance Research Database (NHIRD).

## Materials and methods

### Ethics approval and consent to participate

This study was approved by the Institutional Review Board of Chung Shan Medical University Hospital (Approval number CS15134) in Taiwan. The requirement for written consent from study subjects was waived by the Institutional Review Board, as the LHID consists of de-identified secondary data.

### Data source and study population

This retrospective cohort study was conducted by using data from the National Health Insurance Research Database (NHIRD), a database covering 99% of Taiwan's population of 23 million beneficiaries. This database includes all insurance claims data, such as outpatient visits, emergency visits, and hospitalizations. One million subjects (Longitudinal Health Insurance Database, LHID) [28–30] were randomly sampled from the 23 million beneficiaries, providing data between 1999 and 2013. This sampled database was de-identified in accordance with privacy protocols. The Institutional Review Board of Chung Shan Medical University Hospital (Approval number CS15134) has approved this study.

The study population comprised patients with newly diagnosed constipation (ICD-9-CM codes = 564.0) from 2000 to 2010. To ensure the accuracy of diagnoses, only patients with at least three outpatient visits or one hospitalization were included. The index date of this cohort was set as the date of first diagnosis of constipation. In order to ensure that only new-onset subjects were enrolled, patients diagnosed with allergic rhinitis (ICD-9-CM = 477) before the index date were excluded. The non-constipation group was composed of patients who had never been diagnosed with constipation (ICD-9-CM = 564.0) for the period 1999 to 2013. The index date for the non-constipation group was determined according to the respective matched cases. The flowchart of enrolment is depicted in Fig 1.

The outcome variable was defined as a diagnosis of allergic rhinitis (ICD-9-CM = 477) with at least three outpatient visits or one hospitalization. This study was followed up till the occurrence of AR, 31 December 2013, or until their records were censored for death, emigration, or discontinuation of enrolment in the National Health Insurance system.

### Covariates and matching

The baseline characteristics were age, gender, related comorbidities including hyperlipidemia (ICD-9-CM = 272.0–272.4), hypertension (ICD-9-CM = 401–405), chronic kidney disease (ICD-9-CM = 585), chronic liver disease (ICD-9-CM = 571), diabetes (ICD-9-CM = 250), chronic obstructive pulmonary disease (COPD) (ICD-9-CM = 491, 492, 496), autoimmune diseases such as systemic lupus erythematosus (SLE) (ICD-9-CM = 710.0), rheumatoid arthritis (RA) (ICD-9-CM = 714.0), Sjogren's syndrome (ICD-9-CM = 710.2), ankylosing spondylitis (AS) (ICD-9-CM = 720.0), and the diseases predisposing constipation [31], including dyspepsia (ICD-9-CM = 536.8), gastroesophageal reflux disease(GERD) (ICD-9-CM = 530.11, 530.8x), irritable bowel syndrome (IBS) (ICD-9-CM = 564.1), gastrointestinal tract cancers

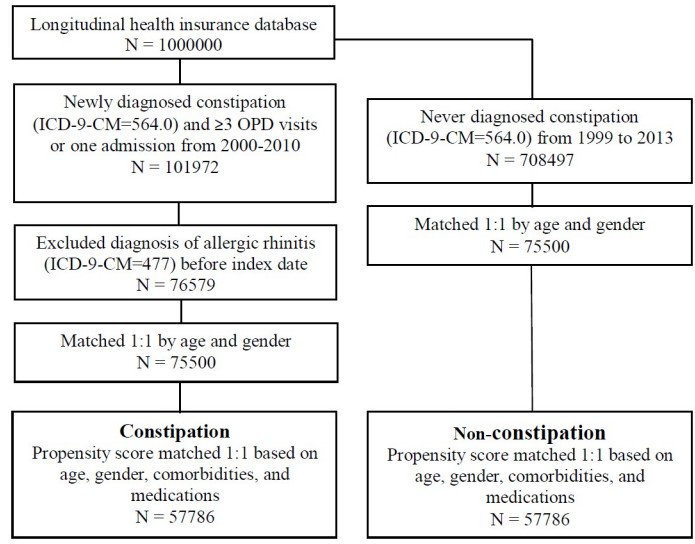

**Fig 1. Flowchart of study.**

(ICD-9-CM = 150–159), colonic polyp (ICD-9-CM = V12.72, 211.3 and 211.4), anxiety (ICD-9-CM = 300.0), depression (ICD-9-CM = 296.2, 296.3, 300.4 and 311), hypothyroidism (ICD-9-CM = 243, 244), Parkinson's disease (ICD-9-CM = 322), multiple sclerosis (ICD-9-CM = 340), spinal cord injury (ICD-9-CM = 806, 952). The comorbidities were defined as occurring within one year prior to the index date and with at least three outpatient visits or one hospitalization. In addition, medications containing corticosteroids, antihistamines, non-steroidal anti-inflammatory drugs, calcium channel blockers, diuretics, opioids, antidepressants, serotonin antagonists, anticonvulsants, antispasmodic, iron supplement, calcium supplement during the study period were included and defined as usage for $\geq$30 days.

Propensity-score matching was performed to match the two groups based on age, gender, hypertension, hyperlipidemia, chronic liver disease, diabetes, chronic kidney disease, COPD, SLE, RA, Sjogren's syndrome, AS, dyspepsia, GERD, IBS, gastrointestinal tract cancers, colonic polyp, anxiety, depression, hypothyroidism, Parkinson's disease, multiple sclerosis, spinal cord injury, and various medications. The propensity score was the probability estimated by logistic regression, with the binary variable being whether or not patients had constipation, i.e., constipation vs. non-constipation groups. Propensity-score matching was performed in order to balance the heterogeneity of the two groups [32].

## Statistical analysis

The constipation group and non-constipation group were compared by Chi-square test or Fisher's exact test for categorical variables and Student's t- test for continuous variables. Moreover, we also used absolute standardized differences (ASD) to perform the difference between the two groups. When the absolute standardized difference was less than 0.1, the characteristics of both groups were considered to be similar [33]. Kaplan-Meier analysis was used to calculate the cumulative incidence of AR from the index date and log-rank test was used to test the significance. Cox proportional hazard model was used to estimate the hazard ratio of AR between the constipation group and non-constipation group. SPSS version 18.0 (SPSS Inc., Chicago, IL, USA) was used for all statistical analyses.

## Results

We identified 57786 patients with constipation and 57786 matched controls between 1999 and 2013 from the LHID, a subset of Taiwanese NHIRD. Table 1 shows the demographic characteristics of the patients. The constipation and non-constipation groups were similar in age and gender distribution; however, women had twice the incidence of constipation compared with men. Also, there were no statistically significant differences between the constipation cohort and the non-constipation cohort after propensity score matching. Table 2 displays the incidence density and risk factors for AR. The incidence of AR was 32.2 per 1,000 person-years in constipation patients, which was higher than the rate of 14.8 per 1,000 person-years found in non-constipation patients. After adjustment, patients with constipation had a significantly higher risk of AR than those without constipation (aHR, 2.30; 95% CI, 2.23–2.37; P < 0.001), and older age groups were associated with a lower risk of developing AR when compared with the <20 years age group. Compared with women, men had a higher risk of AR (aHR, 1.07; 95% CI, 1.04–1.10; P < 0.001). In terms of comorbidities, patients with hypertension(aHR, 1.61; 95% CI, 1.52–1.71; P < 0.001), hyperlipidemia (aHR, 1.23; 95% CI, 1.13–1.34; P < 0.001), chronic liver disease (aHR, 1.44; 95% CI, 1.31–1.58; P < 0.001), COPD (aHR, 2.02; 95% CI, 1.84–2.21; P < 0.001), RA (aHR, 1.58; 95% CI, 1.18–2.12; P = 0.002), AS (aHR, 1.81; 95% CI, 1.07–3.06; P = 0.027), SLE (aHR, 1.88; 95% CI, 1.25–2.83; P = 0.002), Sjogren's syndrome (aHR, 2.45; 95% CI, 1.68–3.58; P< 0.001), dyspepsia (aHR, 1.34; 95% CI, 1.18–1.52; P < 0.001), IBS (aHR, 1.45; 95% CI, 1.21–1.73; P < 0.001), anxiety (aHR, 1.69; 95% CI, 1.52–1.88; P < 0.001) and depression (aHR, 1.41; 95% CI, 1.22–1.63; P < 0.001) were at higher risk of AR. In Table 3, subgroup analyses were performed to assess the relationship between constipation and AR based on demographic characteristics. In patients aged 20–39 years, compared with the non-constipation cohort, there was a 2.24-fold higher risk of AR in the constipation cohort (aHR; 95% CI, 2.12–2.36; P < 0.001). Patients aged <20, 40–64, and ≥65 years had a 2.09, 2.05, and 2.07-fold risk of AR in the constipation cohort (aHR; 95% CI, 1.98–2.20, 1.94–2.18 and 1.92–2.23; P < 0.001), respectively. Among women, compared with patients without constipation, there was a 2.13-fold higher risk of AR in patients with constipation (aHR; 95% CI, 2.05–2.20; P < 0.001). Among men, there was 2.02-fold higher risk of AR in patients with constipation (aHR; 95% CI, 1.92–2.13; P < 0.001). Furthermore, patients with constipation had a significantly higher likelihood of AR, regardless of sex, and with or without comorbidities including hyperlipidemia, hypertension, chronic kidney disease, chronic liver disease, diabetes, COPD, RA, dyspepsia, IBS, and anxiety. The interaction effect was to compare the hazard ratios between different subgroups. In subgroup of non-hypertension, non-hyperlipidemia, non-chronic liver disease, and non-depression, constipation group had significant higher risk of allergic rhinitis. The Kaplan–Meier curves are shown in Fig 2. The cumulative incidence of AR was significantly higher in constipated patients than non-constipated patients, and the log-rank test for the comparison of cumulative incidence curves resulted in a P-value of <0.001.

## Discussion

In current study, we found a 2.3-fold higher risk of incidental AR in constipated patients than in non-constipated patients. To date and to the best of our knowledge, this is the largest and first cohort study to use a longitudinal nationwide population-based dataset to identify an increased AR risk among patients with constipation. This association may have important clinical and pathophysiological implications. Our findings highlight the considerably higher risk of AR in constipated patients. Constipation seems to be influential in development of AR. Clinicians should therefore keep a higher index of suspicion for AR in constipated patients.

**Table 1. Demographic characteristics of constipation group and non-constipation group.**

| | Before propensity score matched | | | | | | After propensity score matched | | | | | |
| | Constipation (N = 75500) | | Non-constipation (N = 75500) | | | | Constipation (N = 57786) | | Non-constipation (N = 57786) | | | |
| | n | % | n | % | ASD | p-value | n | % | N | % | ASD | p-value |
|---|---|---|---|---|---|---|---|---|---|---|---|---|
| Age | | | | | <0.001 | 1 | | | | | 0.026 | 0.002 |
| <20 | 11001 | 14.6 | 11001 | 14.6 | | | 9722 | 16.8 | 10059 | 17.4 | | |
| 20–39 | 20395 | 27.0 | 20395 | 27.0 | | | 16677 | 28.9 | 16274 | 28.2 | | |
| 40–64 | 22337 | 29.6 | 22337 | 29.6 | | | 16194 | 28.0 | 16484 | 28.5 | | |
| ≥65 | 21767 | 28.8 | 21767 | 28.8 | | | 15193 | 26.3 | 14969 | 25.9 | | |
| Mean ± SD | 46.4 ± 23.6 | | 46.4 ± 23.6 | | <0.001 | 1 | 44.4 ± 24 | | 44 ± 23.8 | | 0.018 | 0.002 |
| Gender | | | | | <0.001 | 1 | | | | | 0.003 | 0.569 |
| Female | 49243 | 65.2 | 49243 | 65.2 | | | 38615 | 66.8 | 38706 | 67.0 | | |
| Male | 26257 | 34.8 | 26257 | 34.8 | | | 19171 | 33.2 | 19080 | 33.0 | | |
| Hypertension | 15253 | 20.2 | 10458 | 13.9 | 0.170 | <0.001 | 9082 | 15.7 | 9043 | 15.6 | 0.002 | 0.752 |
| Hyperlipidemia | 3938 | 5.2 | 2615 | 3.5 | 0.086 | <0.001 | 2275 | 3.9 | 2256 | 3.9 | 0.002 | 0.773 |
| Chronic liver disease | 3058 | 4.1 | 1497 | 2.0 | 0.121 | <0.001 | 1465 | 2.5 | 1407 | 2.4 | 0.006 | 0.273 |
| Chronic kidney disease | 883 | 1.2 | 493 | 0.7 | 0.054 | <0.001 | 435 | 0.8 | 415 | 0.7 | 0.004 | 0.491 |
| Diabetes | 7808 | 10.3 | 4387 | 5.8 | 0.167 | <0.001 | 4085 | 7.1 | 4020 | 7.0 | 0.004 | 0.454 |
| COPD | 3128 | 4.1 | 1573 | 2.1 | 0.119 | <0.001 | 1491 | 2.6 | 1484 | 2.6 | 0.001 | 0.897 |
| Rheumatoid arthritis | 281 | 0.4 | 165 | 0.2 | 0.028 | <0.001 | 148 | 0.3 | 143 | 0.2 | 0.002 | 0.769 |
| Ankylosing spondylitis | 92 | 0.1 | 34 | 0.05 | 0.027 | <0.001 | 37 | 0.1 | 34 | 0.1 | 0.002 | 0.722 |
| SLE | 75 | 0.1 | 60 | 0.1 | 0.007 | 0.197 | 52 | 0.1 | 45 | 0.1 | 0.004 | 0.477 |
| Sjogren's syndrome | 124 | 0.2 | 65 | 0.1 | 0.022 | <0.001 | 55 | 0.1 | 61 | 0.1 | 0.003 | 0.577 |
| Dyspepsia | 1867 | 2.5 | 567 | 0.8 | 0.137 | <0.001 | 615 | 1.1 | 566 | 1.0 | 0.008 | 0.152 |
| GERD | 528 | 0.7 | 127 | 0.2 | 0.081 | <0.001 | 145 | 0.3 | 127 | 0.2 | 0.006 | 0.275 |
| Irritable bowel syndrome | 955 | 1.3 | 281 | 0.4 | 0.099 | <0.001 | 321 | 0.6 | 281 | 0.5 | 0.010 | 0.102 |
| Gastrointestinal tract cancer | 905 | 1.2 | 502 | 0.7 | 0.056 | <0.001 | 489 | 0.8 | 460 | 0.8 | 0.006 | 0.345 |
| Colonic polyp | 170 | 0.2 | 53 | 0.1 | 0.040 | <0.001 | 66 | 0.1 | 53 | 0.1 | 0.007 | 0.233 |
| Anxiety | 2566 | 3.4 | 956 | 1.3 | 0.142 | <0.001 | 984 | 1.7 | 931 | 1.6 | 0.007 | 0.222 |
| Depression | 1988 | 2.6 | 534 | 0.7 | 0.151 | <0.001 | 584 | 1.0 | 527 | 0.9 | 0.010 | 0.086 |
| Hypothyroidism | 213 | 0.3 | 117 | 0.2 | 0.027 | <0.001 | 122 | 0.2 | 110 | 0.2 | 0.005 | 0.430 |
| Parkinson's disease | 879 | 1.2 | 250 | 0.3 | 0.097 | <0.001 | 247 | 0.4 | 250 | 0.4 | 0.001 | 0.893 |
| Multiple sclerosis | 10 | 0.01 | 2 | 0.003 | 0.012 | 0.021 | 1 | 0.002 | 2 | 0.003 | 0.003 | 0.999 |
| Spinal cord injury | 267 | 0.4 | 48 | 0.1 | 0.064 | <0.001 | 55 | 0.1 | 48 | 0.1 | 0.004 | 0.490 |
| Corticosteroids | 20240 | 26.8 | 12612 | 16.7 | 0.247 | <0.001 | 11729 | 20.3 | 11673 | 20.2 | 0.002 | 0.682 |
| Antihistamines | 51295 | 67.9 | 38014 | 50.3 | 0.364 | <0.001 | 35663 | 61.7 | 35677 | 61.7 | 0.000 | 0.932 |
| NSAIDs | 55459 | 73.5 | 43706 | 57.9 | 0.332 | <0.001 | 39317 | 68.0 | 39457 | 68.3 | 0.005 | 0.377 |
| Calcium channel blockers | 21533 | 28.5 | 16949 | 22.4 | 0.140 | <0.001 | 13656 | 23.6 | 13556 | 23.5 | 0.004 | 0.488 |
| Diuretics | 18886 | 25.0 | 12421 | 16.5 | 0.212 | <0.001 | 10842 | 18.8 | 10868 | 18.8 | 0.001 | 0.845 |
| Opioids | 9228 | 12.2 | 4973 | 6.6 | 0.194 | <0.001 | 4657 | 8.1 | 4583 | 7.9 | 0.005 | 0.422 |
| Antidepressants | 14216 | 18.8 | 5566 | 7.4 | 0.345 | <0.001 | 5698 | 9.9 | 5455 | 9.4 | 0.014 | 0.015 |
| Serotonin (5HT3) antagonists | 772 | 1.0 | 472 | 0.6 | 0.044 | <0.001 | 428 | 0.7 | 444 | 0.8 | 0.003 | 0.587 |
| Anticonvulsants | 11968 | 15.9 | 4833 | 6.4 | 0.304 | <0.001 | 4910 | 8.5 | 4681 | 8.1 | 0.014 | 0.015 |
| Antispasmodic | 19669 | 26.1 | 11406 | 15.1 | 0.273 | <0.001 | 10934 | 18.9 | 10894 | 18.9 | 0.002 | 0.764 |
| Iron supplement | 4229 | 5.6 | 2365 | 3.1 | 0.121 | <0.001 | 2212 | 3.8 | 2176 | 3.8 | 0.003 | 0.580 |

(*Continued*)

**Table 1.** (Continued)

| | Before propensity score matched | | | | | | After propensity score matched | | | | | |
| | Constipation | | Non-constipation | | | | Constipation | | Non-constipation | | | |
| | (N = 75500) | | (N = 75500) | | | | (N = 57786) | | (N = 57786) | | | |
| | n | % | n | % | ASD | p-value | n | % | N | % | ASD | p-value |
| Calcium supplement | 7654 | 10.1 | 4677 | 6.2 | 0.144 | <0.001 | 4186 | 7.2 | 4153 | 7.2 | 0.002 | 0.708 |

ASD: Absolute standardized differences.

COPD: Chronic obstructive pulmonary disease.

SLE: Systemic lupus erythematosus.

GERD: Gastroesophageal reflux disease.

NSAIDs: Non-steroidal anti-inflammatory drugs.

On the same note, constipated patients should be informed of the possible risk of AR. We also suggest that people need to maintain good bowel habits to avoid constipation that might contribute to AR.

Our findings were in line with an epidemiology study using questionnaire survey in Japan [11], including 21802 of senior high school students, aged from 15 to 18 years old, analyzing the relevant risk factors for development and remission of atopic disease. Results from this study indicated that constipated students had 1.17-fold risk for AR. However, the design of the study might have allowed some residual bias to exist and might not have controlled for possible confounding factors with propensity score methods. Similar findings were found in our study for patients under the age of 20 with a 2.09-fold AR risk. Moreover, we observed a significant higher risk of AR not only during childhood, but throughout adulthood. Consistent with our finding that there was a significant correlation between constipation and allergic diseases, Palmieri, M. et al. found a significant difference in the prevalence of atopic diseases proven by skin prick tests between the constipated children and the control group (17/52 = 33% versus 11/74 = 15%; p = 0.03) [34]. In addition, Jones et al. demonstrated that the overlap of atopy among 23471 functional gastrointestinal disorders patients, and the risk of rhinitis was 1.66-fold higher in patients with constipation than controls [8]. We also noted that hyperlipidemia, hypertension, chronic kidney disease, chronic liver disease, diabetes, COPD, RA, dyspepsia, IBS, and anxiety were associated with greater risk of AR in patients with constipation. RA was a comorbidity with a relatively higher AR risk (HR: 2.68, 95% CI, 1.41–5.1) compared with other comorbidities. Both RA and AR are characterized by the regulatory T-cells dysfunction [35]. Besides, constipation may worsen pre-existing dysbiosis in patients with RA [36, 37]. Moreover, patients with autoimmunity appeared to be predisposed to subsequent AR in our study. However, the risk for AR of other autoimmune diseases such as SLE, AS, Sjogren's syndrome did not reach statistical significance in subgroup analysis. There might have been insufficient statistical power to detect significant differences due to the low incidence of these autoimmune diseases in our patient population.

The pathophysiology underlying the relationship between constipation and subsequent AR remains unclear. In recent times, it was shown that the gut microbiota has a pivotal role in the modulation of immunity [38–40]. Khalif et al. [17] and Feng et al. [41] reported that patients with constipation had a decreased abundance in Bifidobacteria and Lactobacillus. Khalif et al. observed that constipation was correlated with significant alterations in the fecal flora, gut permeability (leaky gut), and immune response, and that alleviation of constipation tended to reverse these changes [17]. Intestinal conditions such as constipation could affect immunity by altering gut microflora and permeability, leading to hyper-secretion of proinflammatory or

**Table 2. Multiple Cox proportional hazard regression for the estimation of adjusted hazard ratios for AR.**

| | No. of allergic rhinitis | Observed Person-Years | ID | Crude HR | 95% C.I. | p value | Adjusted HR[†] | 95% C.I. | p value |
|---|---|---|---|---|---|---|---|---|---|
| Group | | | | | | | | | |
| Non-constipation | 7104 | 480068 | 14.8 | 1 | | | 1 | | |
| Constipation | 13076 | 405708 | 32.2 | 2.09 | 2.03–2.15 | <0.001 | 2.30 | 2.23–2.37 | <0.001 |
| Age | | | | | | <0.001 | | | <0.001 |
| <20 | 6097 | 163913 | 37.2 | 1 | | | 1 | | |
| 20–39 | 5952 | 270959 | 22.0 | 0.57 | 0.55–0.59 | <0.001 | 0.58 | 0.56–0.6 | <0.001 |
| 40–64 | 4924 | 262465 | 18.8 | 0.48 | 0.47–0.5 | <0.001 | 0.62 | 0.6–0.65 | <0.001 |
| ≥65 | 3207 | 188439 | 17.0 | 0.41 | 0.4–0.43 | <0.001 | 0.65 | 0.62–0.69 | <0.001 |
| Gender | | | | | | | | | |
| Female | 13787 | 617078 | 22.3 | 1 | | | 1 | | |
| Male | 6393 | 268699 | 23.8 | 1.04 | 1.01–1.07 | 0.022 | 1.07 | 1.04–1.1 | <0.001 |
| Hypertension | 2219 | 113499 | 19.6 | 0.78 | 0.75–0.82 | <0.001 | 1.61 | 1.52–1.71 | <0.001 |
| Hyperlipidemia | 642 | 28879 | 22.2 | 0.91 | 0.84–0.98 | 0.014 | 1.23 | 1.13–1.34 | <0.001 |
| Chronic liver disease | 466 | 17869 | 26.1 | 1.10 | 1–1.2 | 0.048 | 1.44 | 1.31–1.58 | <0.001 |
| Chronic kidney disease | 69 | 3696 | 18.7 | 0.71 | 0.56–0.9 | 0.004 | 0.95 | 0.75–1.21 | 0.678 |
| Diabetes | 822 | 47744 | 17.2 | 0.69 | 0.64–0.73 | <0.001 | 0.86 | 0.8–0.93 | <0.001 |
| COPD | 528 | 13425 | 39.3 | 1.55 | 1.42–1.69 | <0.001 | 2.02 | 1.84–2.21 | <0.001 |
| Rheumatoid arthritis | 46 | 1887 | 24.4 | 1.04 | 0.78–1.39 | 0.783 | 1.58 | 1.18–2.12 | 0.002 |
| Ankylosing spondylitis | 14 | 475 | 29.5 | 1.24 | 0.74–2.1 | 0.412 | 1.81 | 1.07–3.06 | 0.027 |
| SLE | 24 | 649 | 37.0 | 1.55 | 1.04–2.32 | 0.031 | 1.88 | 1.25–2.83 | 0.002 |
| Sjogren's syndrome | 28 | 653 | 42.9 | 1.71 | 1.18–2.48 | 0.005 | 2.45 | 1.68–3.58 | <0.001 |
| Dyspepsia | 244 | 7542 | 32.4 | 1.37 | 1.21–1.55 | <0.001 | 1.34 | 1.18–1.52 | <0.001 |
| GERD | 24 | 1283 | 18.7 | 0.68 | 0.45–1.01 | 0.056 | 0.59 | 0.4–0.88 | 0.010 |
| Irritable bowel syndrome | 121 | 3789 | 31.9 | 1.35 | 1.13–1.61 | 0.001 | 1.45 | 1.21–1.73 | <0.001 |
| Gastrointestinal tract cancers | 61 | 3840 | 15.9 | 0.62 | 0.48–0.8 | <0.001 | 0.78 | 0.61–1.01 | 0.061 |
| Colonic polyp | 19 | 614 | 31.0 | 1.23 | 0.78–1.93 | 0.369 | 1.46 | 0.93–2.29 | 0.102 |
| Anxiety | 377 | 12325 | 30.6 | 1.27 | 1.15–1.41 | <0.001 | 1.69 | 1.52–1.88 | <0.001 |
| Depression | 195 | 7018 | 27.8 | 1.15 | 1–1.33 | 0.049 | 1.41 | 1.22–1.63 | <0.001 |
| Hypothyroidism | 38 | 1554 | 24.5 | 1.02 | 0.74–1.41 | 0.884 | 1.07 | 0.78–1.47 | 0.688 |
| Parkinson's disease | 29 | 2284 | 12.7 | 0.48 | 0.34–0.7 | <0.001 | 0.49 | 0.34–0.71 | <0.001 |
| Spinal cord injury | 10 | 627 | 16.0 | 0.66 | 0.36–1.23 | 0.192 | 0.83 | 0.44–1.54 | 0.547 |

COPD: Chronic obstructive pulmonary disease.

SLE: Systemic Lupus Erythematosus.

GERD: Gastroesophageal reflux disease.

ID: Incidence density (per 1000 person-years).

[†]Multiple Cox proportional hazard regression was used to adjust for age, gender, comorbidities, and medications.

inflammatory biomarkers, such as chemokines and cytokines [39, 42]. Cirali et al. found that neopterin, IL-6, and IL-12 and levels of constipated children were higher than in the non-constipation group [43], indicating that subclinical inflammation existed in patients with constipation. Mokhtare et al. also showed higher levels of TNF-α, IL-1, and IL-6in geriatric patients with chronic constipation, compared with healthy controls [42]. In experiments, TNF-α was required to produce antigen-specific IgE and to induce T-helper 2 cytokines and chemokines in allergic rhinitis [44, 45]. Moreover, gut microbiota-derived metabolites (including short-chain fatty acids such as acetate, butyrate, and propionate.) produced from high-fiber diets have been implicated as protective against allergy [46]. Trompette et al. [47] showed that mice

**Table 3. Subgroup analysis of hazard ratios (95% CI) of AR for patients with and without constipation by age, gender, and comorbidities.**

| | Constipation | | Non-constipation | | HR | 95% C.I. | p value |
|---|---|---|---|---|---|---|---|
| | N | No. of allergic rhinitis | N | No. of allergic rhinitis | | | |
| Age | | | | | | | |
| <20 | 9722 | 3851 | 10059 | 2246 | 2.09 | 1.98–2.2 | <0.001 |
| 20–39 | 16677 | 3999 | 16274 | 1953 | 2.24 | 2.12–2.36 | <0.001 |
| 40–64 | 16194 | 3130 | 16484 | 1794 | 2.05 | 1.94–2.18 | <0.001 |
| ≥65 | 15193 | 2096 | 14969 | 1111 | 2.07 | 1.92–2.23 | <0.001 |
| | | | | | | | p† = 0.085 |
| Gender | | | | | | | |
| Female | 38615 | 8994 | 38706 | 4793 | 2.13 | 2.05–2.2 | <0.001 |
| Male | 19171 | 4082 | 19080 | 2311 | 2.02 | 1.92–2.13 | <0.001 |
| | | | | | | | p† = 0.154 |
| Hypertension | | | | | | | |
| No | 48704 | 11684 | 48743 | 6277 | 2.13 | 2.06–2.19 | <0.001 |
| Yes | 9082 | 1392 | 9043 | 827 | 1.85 | 1.7–2.02 | <0.001 |
| | | | | | | | p† = 0.003 |
| Hyperlipidemia | | | | | | | |
| No | 55511 | 12682 | 55530 | 6856 | 2.11 | 2.04–2.17 | <0.001 |
| Yes | 2275 | 394 | 2256 | 248 | 1.76 | 1.5–2.07 | <0.001 |
| | | | | | | | p† = 0.037 |
| Chronic liver disease | | | | | | | |
| No | 56321 | 12787 | 56379 | 6927 | 2.10 | 2.04–2.17 | <0.001 |
| Yes | 1465 | 289 | 1407 | 177 | 1.70 | 1.41–2.05 | <0.001 |
| | | | | | | | p† = 0.029 |
| Chronic kidney disease | | | | | | | |
| No | 57351 | 13032 | 57371 | 7079 | 2.09 | 2.03–2.16 | <0.001 |
| Yes | 435 | 44 | 415 | 25 | 1.77 | 1.08–2.89 | 0.023 |
| | | | | | | | p† = 0.506 |
| Diabetes | | | | | | | |
| No | 53701 | 12532 | 53766 | 6826 | 2.10 | 2.04–2.16 | <0.001 |
| Yes | 4085 | 544 | 4020 | 278 | 2.07 | 1.79–2.39 | <0.001 |
| | | | | | | | p† = 0.868 |
| Chronic obstructive pulmonary disease | | | | | | | |
| No | 56295 | 12746 | 56302 | 6906 | 2.10 | 2.04–2.16 | <0.001 |
| Yes | 1491 | 330 | 1484 | 198 | 1.78 | 1.49–2.12 | <0.001 |
| | | | | | | | p† = 0.086 |
| Rheumatoid arthritis | | | | | | | |
| No | 57638 | 13043 | 57643 | 7091 | 2.09 | 2.03–2.15 | <0.001 |
| Yes | 148 | 33 | 143 | 13 | 2.68 | 1.41–5.1 | 0.003 |
| | | | | | | | p† = 0.443 |
| Ankylosing spondylitis | | | | | | | |
| No | 57749 | 13069 | 57752 | 7097 | 2.09 | 2.03–2.16 | <0.001 |
| Yes | 37 | 7 | 34 | 7 | 0.87 | 0.31–2.49 | 0.799 |
| | | | | | | | p† = 0.097 |
| Systemic lupus erythematosus | | | | | | | |
| No | 57734 | 13062 | 57741 | 7094 | 2.09 | 2.03–2.16 | <0.001 |
| Yes | 52 | 14 | 45 | 10 | 1.27 | 0.56–2.87 | 0.561 |
| | | | | | | | p† = 0.233 |

*(Continued)*

**Table 3.** (Continued)

| | Constipation | | Non-constipation | | | | |
|---|---|---|---|---|---|---|---|
| | N | No. of allergic rhinitis | N | No. of allergic rhinitis | HR | 95% C.I. | p value |
| Sjogren's syndrome | | | | | | | |
| No | 57731 | 13058 | 57725 | 7094 | 2.09 | 2.03–2.15 | <0.001 |
| Yes | 55 | 18 | 61 | 10 | 2.11 | 0.97–4.57 | 0.059 |
| | | | | | | | p† = 0.924 |
| Dyspepsia | | | | | | | |
| No | 57171 | 12912 | 57220 | 7024 | 2.09 | 2.03–2.16 | <0.001 |
| Yes | 615 | 164 | 566 | 80 | 2.03 | 1.55–2.65 | <0.001 |
| | | | | | | | p† = 0.882 |
| GERD | | | | | | | |
| No | 57641 | 13061 | 57659 | 7095 | 2.09 | 2.03–2.16 | <0.001 |
| Yes | 145 | 15 | 127 | 9 | 1.53 | 0.67–3.5 | 0.313 |
| | | | | | | | p† = 0.454 |
| Irritable bowel syndrome | | | | | | | |
| No | 57465 | 12997 | 57505 | 7062 | 2.10 | 2.04–2.16 | <0.001 |
| Yes | 321 | 79 | 281 | 42 | 1.70 | 1.17–2.47 | 0.006 |
| | | | | | | | p† = 0.308 |
| Gastrointestinal tract cancer | | | | | | | |
| No | 57297 | 13042 | 57326 | 7077 | 2.10 | 2.04–2.16 | <0.001 |
| Yes | 489 | 34 | 460 | 27 | 1.40 | 0.85–2.33 | 0.189 |
| | | | | | | | p† = 0.113 |
| Colonic polyp | | | | | | | |
| No | 57720 | 13063 | 57733 | 7098 | 2.09 | 2.03–2.15 | <0.001 |
| Yes | 66 | 13 | 53 | 6 | 1.78 | 0.68–4.68 | 0.244 |
| | | | | | | | p† = 0.738 |
| Anxiety | | | | | | | |
| No | 56802 | 12826 | 56855 | 6977 | 2.09 | 2.03–2.16 | <0.001 |
| Yes | 984 | 250 | 931 | 127 | 2.08 | 1.68–2.57 | <0.001 |
| | | | | | | | p† = 0.974 |
| Depression | | | | | | | |
| No | 57202 | 12963 | 57259 | 7022 | 2.10 | 2.04–2.16 | <0.001 |
| Yes | 584 | 113 | 527 | 82 | 1.31 | 0.99–1.75 | 0.059 |
| | | | | | | | p† = 0.001 |
| Hypothyroidism | | | | | | | |
| No | 57664 | 13054 | 57676 | 7088 | 2.09 | 2.04–2.16 | <0.001 |
| Yes | 122 | 22 | 110 | 16 | 1.36 | 0.72–2.6 | 0.346 |
| | | | | | | | p† = 0.228 |
| Parkinson's disease | | | | | | | |
| No | 57539 | 13060 | 57536 | 7091 | 2.10 | 2.04–2.16 | <0.001 |
| Yes | 247 | 16 | 250 | 13 | 1.14 | 0.55–2.36 | 0.733 |
| | | | | | | | p† = 0.104 |
| Spinal cord injury | | | | | | | |
| No | 57731 | 13068 | 57738 | 7102 | 2.09 | 2.03–2.15 | <0.001 |
| Yes | 55 | 8 | 48 | 2 | 3.91 | 0.83–18.46 | 0.085 |
| | | | | | | | p† = 0.467 |

†p for difference of HR between subgroup.

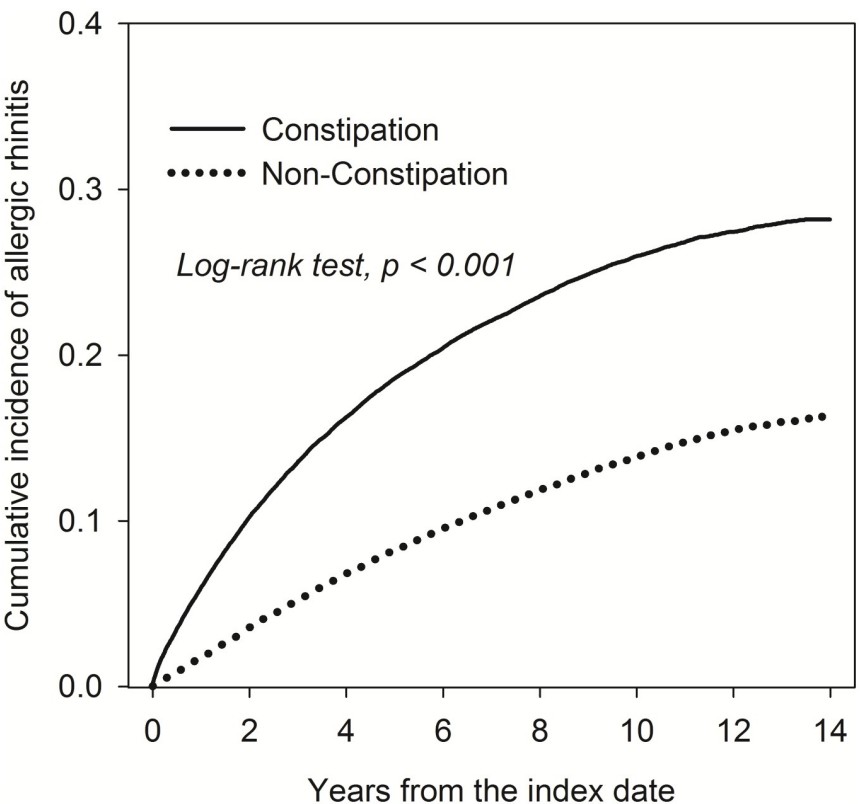

**Fig 2. Kaplan–Meier curves of the cumulative probability of AR in the study groups.**

fed low-fiber diets before nasal exposure to house dust mite extract had increased IL-4, IL-5, IL-13, and IL-17A in airway tissue, increased goblet cell hyperplasia and mucus secretion, and higher IgE levels in the serum. By contrast, mice fed high-fiber diets had a normal mucin secretion and lower cytokine levels. Moreover, fiber intake affected the intestinal microbiome composition with relative abundance of Bifidobacteriaceae and Bacteroidaceae species [47]. Low fiber intake in constipated subjects might also play a role in atopy development. Although Parthasarathy et al. have offered that mucosal microbiota analysis could discriminate 25 constipated adults from healthy controls with 94% accuracy [48]. At variance, a recent study mentioned that no disease specific separation was observed by PCoA and by calculation of diversity indices in constipated children and healthy matched controls [19]. However, both groups could be discriminated with 82% accuracy by ridge regression. Although supportive data from previous studies rarely link constipation to AR, our findings provide support to the hypothesis that constipation and AR might share a similar underlying etiological pathways related to dysbiosis. Nevertheless, the mechanism responsible for constipation-mediated dysbiosis or immune dysfunction, which precipitates AR development, requires further in-depth study.

The major advantages of this big data study were the large sample size and the relatively long time of follow-up, in which a complete history of the medical services used was available for all cases and controls. Therefore, there were minimal selection, recall, and information biases that made testing our hypothesis more feasible. Nonetheless, there were several limitations that should be noted. Firstly, the NHIRD does not disclose information regarding the patients' diet, socioeconomic status, family history, personal lifestyle, such as smoking, alcohol

drinking, and dietary preference, and environmental exposures, which may be associated risk factors for development of AR. Although we adjusted for a variety of comorbidities and medications, and matched propensity scores to reduce the confounders, unmeasured factors might have biased our results. Secondly, the diagnoses of constipation, AR, and comorbidities were entirely dependent on the ICD-9 codes in the administrative dataset. Therefore, the accuracy of diagnoses could not be verified by personal review of medical records and this may have resulted in misclassification. It is worth noting that these misclassifications were more likely to be random, and associations are often underestimated rather than overestimated. However, Taiwan's NHI administration has set up an ad hoc committee to monitor the accuracy of claimed data to prevent violations. Furthermore, we only selected subjects that were repeatedly coded to increase the validity and accuracy of the diagnoses. Thirdly, relevant clinical information, such as laboratory data including cytokines and gene-expression changes, imaging findings, and fecal microbiota assessments were unavailable in the database and therefore could not be included in the analyses.

There is mounting evidence showing that fecal microbiota transplantation (FMT) may be an effective therapeutic approach for intractable constipation [20, 21, 41]. This implies that intestinal dysbiosis is causally related to the pathogenesis or a consequence of constipation. Modulation of the intestinal flora to restore a diverse and balanced microbiome may treat or prevent microbiota-related disease. Furthermore, while constipation can be partially controlled with laxatives, but the disrupted gut microflora may not be completely changed. Therefore, as AR appears to be mediated, at least in part, through the microbiota-gut-airway axis, other therapeutic possibilities for constipation should be considered, such as use of probiotics, prebiotics, synbiotics, and FMT to restore the intestinal flora. Our research provides an observational evidence of an association between constipation and AR. Further research should be conducted to determine how constipation changes the composition of the gut microbiota and the extent to which this affects AR. We also speculate that relief of constipation in patients with AR might be helpful when used in combination with other approaches. State-of-the-art metagenomic and metabolomic analyses of the gut microbiota in constipated patients are needed to better understand their interactions with the immune system.

## Conclusion

The risk of AR in constipated patients is twice the risk in non-constipated patients. It seems that physicians should keep a higher index of suspicion for AR in people with constipation. The patency issue of gut could not be ignored in patients with AR. Further comprehensive basic and clinical research is needed to elucidate the mechanisms underlying these associations.

## Acknowledgments

We thank Chien-Heng Lin, Ph.D., Department of Medical Research, Taichung Veterans General Hospital, Taichung, Taiwan, for his valuable advice of the manuscript. This manuscript was edited by Peter Wilds and Wallace Academic Editing.

## Author Contributions

**Conceptualization:** Meng-Che Wu.

**Data curation:** Yu-Hsun Wang.

**Formal analysis:** Meng-Che Wu, Yu-Hsun Wang.

**Funding acquisition:** James Cheng-Chung Wei.

**Investigation:** Meng-Che Wu, Ming-Shiou Jan, Jeng-Yuan Chiou, Yu-Hsun Wang, James Cheng-Chung Wei.

**Methodology:** Yu-Hsun Wang.

**Project administration:** James Cheng-Chung Wei.

**Resources:** James Cheng-Chung Wei.

**Software:** Yu-Hsun Wang.

**Supervision:** James Cheng-Chung Wei.

**Validation:** Yu-Hsun Wang, James Cheng-Chung Wei.

**Visualization:** Meng-Che Wu.

**Writing – original draft:** Meng-Che Wu, Jeng-Yuan Chiou, Yu-Hsun Wang.

**Writing – review & editing:** Meng-Che Wu, Ming-Shiou Jan, James Cheng-Chung Wei.

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
