## [Editor Report · Decision Letter 0]

20 May 2020

PONE-D-20-09268

Constipation is associated with risk of allergic rhinitis: A nationwide population-based cohort study

PLOS ONE

Dear Dr Wei,

Thank you for submitting your manuscript to PLOS ONE. Before sending out external reviewers, I would like to clarify several points.

There are many factors that are associated with constipation: various GI diseases such as cancers and polyps; imbalance in immune and nervous system function, bile acid metabolism and mucus secretion, and the gastrointestinal microbiota and fermentation (e.g., Nat Rev Dis Primers. 2017 Dec 14;3:17095).

The authors claimed that the high prevalence of constipation is a risk factor for allergic rhinitis. But, any supporting data are not shown to link these two conditions. Just statistical findings cannot prove your hypothesis (maybe coincidental).

In Discussion, the authors described a lot about the results and speculations by other researchers’ studies. However, there are no discussion how these can support your data (i.e., the authors describe others’ speculations to explain your speculations). Please discuss about the presented data.

Also, the background of study participants varies. For example, the authors included autoimmune diseases. Please describe more details (e.g., types of diseases), and explain or speculate how each autoimmune disease can be associated with AR and/or constipation (i.e., pathophysiological mechanisms).

Before sending out for external/statistical review, I would request the authors to present more detail data showing direct, at least suggestive, evidences for your hypothesis (i.e., underlying mechanisms linking constipation and AR). Otherwise, I highly expect that it will be rejected by external reviewers.

We would appreciate receiving your revised manuscript by Jul 04 2020 11:59PM. To enhance the reproducibility of your results, we recommend that if applicable you deposit your laboratory protocols in protocols.io, where a protocol can be assigned its own identifier (DOI) such that it can be cited independently in the future. For instructions see: http://journals.plos.org/plosone/s/submission-guidelines#loc-laboratory-protocols

We look forward to receiving your revised manuscript.

Kind regards,

Tomohiko Ai, M.D., Ph.D.

Academic Editor

PLOS ONE

Journal Requirements:

2. To comply with PLOS ONE submission guidelines, in Table 1, please report your p-values. For more information on PLOS ONE's expectations for statistical reporting, please see " ext-link-type="uri" xlink:type="simple">https://journals.plos.org/plosone/s/submission-guidelines.#loc-statistical-reporting."

Additional Editor Comments (if provided):

There are many factors that are associated with constipation: various GI diseases such as cancers and polyps; imbalance in immune and nervous system function, bile acid metabolism and mucus secretion, and the gastrointestinal microbiota and fermentation (e.g., Nat Rev Dis Primers. 2017 Dec 14;3:17095).

The authors claimed that the high prevalence of constipation is a risk factor for allergic rhinitis. But, any supporting data are not shown. Just statistical findings cannot prove your hypothesis (maybe coincidental).

In Discussion, the authors described a lot about the results and speculations by other researchers’ studies. However, there are no discussion how these can support your data (i.e., the authors describe others’ speculations to explain your speculations). Please discuss about the presented data.

Also, the background of study participants varies. For example, the authors included autoimmune diseases. Please describe more details, and explain or speculate how each autoimmune disease is associated with AR and/or constipation (i.e., pathophysiological mechanisms).

Before sending out for external/statistical review, I would request the authors to present more detail data showing direct, at least suggestive, evidences for your hypothesis (i.e., underlying mechanisms linking constipation and AR). Otherwise, I highly expect that it will be rejected by external reviewers.
---

## [Author Response · Author response to Decision Letter 0]

16 Jul 2020

Tomohiko Ai, M.D., Ph.D.

Academic Editor

Journal: PLOS ONE 

Dear Dr. Tomohiko Ai 

On behalf of my coauthors, I thank the Editor/Reviewers for the comments and am grateful for the opportunity to revise our manuscript (ID: PONE-D-20-09268) titled, “Constipation is associated with risk of allergic rhinitis: A nationwide population-based cohort study”, for consideration of publication as an Original Research in PLOS ONE. Your comments were highly insightful and enabled us to improve the quality of our document. In the following pages are our responses to each comment.

Revisions in the text are shown [yellow highlights]. We have addressed the comments point-by-point, and the corresponding changes have been made in track change in the manuscript, which you will find uploaded alongside this document. We hope that our revisions to the document combined with our accompanying responses will be sufficient to render our document suitable for publication in PLOS ONE.

Kind regards,

James Cheng-Chung Wei, MD, PhD

Institute of Medicine, Chung Shan Medical University 

No. 110, Sec 1, Jianguo N. Road, Taichung, 40201, Taiwan

jccwei@gmail.com

Responses to the Editors/Reviewers’ Comments:

Manuscript ID: PONE-D-20-09268

Manuscript title: Constipation is associated with risk of allergic rhinitis: A nationwide population-based cohort study

Journal: PLOS ONE

Editor/Reviewer(s)' Comments to Author:

1. There are many factors that are associated with constipation: various GI diseases such as cancers and polyps; imbalance in immune and nervous system function, bile acid metabolism and mucus secretion, and the gastrointestinal microbiota and fermentation (e.g., Nat Rev Dis Primers. 2017 Dec 14;3:17095). 

The authors claimed that the high prevalence of constipation is a risk factor for allergic rhinitis. But, any supporting data are not shown to link these two conditions. Just statistical findings cannot prove your hypothesis (maybe coincidental).

Response: 

Thank you for your comments and suggestions. 

We analyzed a variety of comorbid conditions and medications in detail that were associated with constipation based on the reference (Nat Rev Dis Primers. 2017 Dec 14;3:17095). We considered the most chronic gastrointestinal conditions, such as dyspepsia, GERD, IBS, colonic polyps, GI tract cancers. Psychiatric/Mood disorders such as anxiety and depression; autoimmune diseases including SLE, RA, AS, Sjogren’s syndrome; neurological diseases such as Parkinson’s disease, multiple sclerosis, spinal cord injury and hypothyroidism were also enrolled for analysis. In addition, we further considered various medications associated with constipation including non-steroidal anti-inflammatory drugs, calcium channel blockers, diuretics, opioids, antidepressants, serotonin antagonists, anticonvulsants, antispasmodic, iron supplement, calcium supplement. Furthermore, we attempted to use propensity score matching to achieve more complete control of the potential confounders.

We added this paragraph for explanation as below in method section:

The baseline characteristics were age, gender, related comorbidities including hyperlipidemia (ICD-9-CM=272.0-272.4), hypertension (ICD-9-CM=401-405), chronic kidney disease (ICD-9-CM=585), chronic liver disease (ICD-9-CM=571), diabetes (ICD-9-CM=250), chronic obstructive pulmonary disease (COPD) (ICD-9-CM =491, 492, 496), autoimmune diseases such as systemic lupus erythematosus (SLE) (ICD-9-CM=710.0), rheumatoid arthritis (RA) (ICD-9-CM=714.0), Sjogren’s syndrome (ICD-9-CM=710.2), ankylosing spondylitis (AS) (ICD-9-CM=720.0), and the diseases predisposing constipation [31], including dyspepsia (ICD-9-CM=536.8), gastroesophageal reflux disease (GERD) (ICD-9-CM=530.11, 530.8x), irritable bowel syndrome (IBS) (ICD-9-CM=564.1), gastrointestinal tract cancers (ICD-9-CM=150-159), colonic polyp (ICD-9-CM=V12.72, 211.3 and 211.4), anxiety (ICD-9-CM=300.0), depression (ICD-9-CM=296.2, 296.3, 300.4 and 311), hypothyroidism (ICD-9-CM=243, 244), Parkinson’s disease (ICD-9-CM=322), multiple sclerosis (ICD-9-CM=340), spinal cord injury (ICD-9-CM=806, 952). The comorbidities were defined as occurring within one year prior to the index date and with at least three outpatient visits or one hospitalization. In addition, medications containing corticosteroids, antihistamines, non-steroidal anti-inflammatory drugs, calcium channel blockers, diuretics, opioids, antidepressants, serotonin antagonists, anticonvulsants, antispasmodic, iron supplement, calcium supplement during the study period were included and defined as usage for ≥30 days.

 Propensity-score matching was performed to match the two groups based on age, gender, hypertension, hyperlipidemia, chronic liver disease, diabetes, chronic kidney disease, COPD, SLE, RA, Sjogren’s syndrome, AS, dyspepsia, GERD, IBS, gastrointestinal tract cancers, colonic polyp, anxiety, depression, hypothyroidism, Parkinson’s disease, multiple sclerosis, spinal cord injury, and various medications.

We have added new revised tables 1-3 (Attached at the end of the response letter), revised figure 1 and revised figure 2 in the revised manuscript: 

Fig 1. Flowchart of study.

Fig 2. Kaplan–Meier curves of the cumulative probability of AR in the study groups.

We revised in results section as below and also revised in the abstract:

We identified 57786 patients with constipation and 57786 matched controls between 1999 and 2013 from the LHID, a subset of Taiwanese NHIRD………………………..

 …………………………………………………………………………………………Table 2 displays the incidence density and risk factors for AR. The incidence of AR was 32.2 per 1,000 person-years in constipation patients, which was higher than the rate of 14.8 per 1,000 person-years found in non-constipation patients. After adjustment, patients with constipation had a significantly higher risk of AR than those without constipation (aHR, 2.30; 95% CI, 2.23-2.37; P 0.001), and older age groups were associated with a lower risk of developing AR when compared with the 20 years age group. Compared with women, men had a higher risk of AR (aHR, 1.07; 95% CI, 1.04-1.10; P 0.001). In terms of comorbidities, patients with hypertension (aHR, 1.61; 95% CI, 1.52-1.71; P 0.001), hyperlipidemia (aHR, 1.23; 95% CI, 1.13-1.34; P 0.001), chronic liver disease (aHR, 1.44; 95% CI, 1.31-1.58; P 0.001), COPD (aHR, 2.02; 95% CI, 1.84-2.21; P 0.001), RA (aHR, 1.58; 95% CI, 1.18-2.12; P =0.002), AS (aHR, 1.81; 95% CI, 1.07-3.06; P=0.027), SLE (aHR, 1.88; 95% CI, 1.25-2.83; P=0.002), Sjogren’s syndrome (aHR, 2.45; 95% CI, 1.68-3.58; P 0.001), dyspepsia (aHR, 1.34; 95% CI, 1.18-1.52; P 0.001), IBS (aHR, 1.45; 95% CI, 1.21-1.73; P 0.001), anxiety (aHR, 1.69; 95% CI, 1.52-1.88; P 0.001) and depression (aHR, 1.41; 95% CI, 1.22-1.63; P 0.001) were at higher risk of AR. In Table 3, stratified analyses were performed to assess the relationship between constipation and AR based on demographic characteristics. In patients aged 20-39 years, compared with the non-constipation cohort, there was a 2.24-fold higher risk of AR in the constipation cohort (aHR; 95% CI, 2.12-2.36; P 0.001). Patients aged 20, 40-64, and ≥65 years had a 2.09, 2.05, and 2.07-fold risk of AR in the constipation cohort (aHR; 95% CI, 1.98-2.20, 1.94-2.18 and 1.92-2.23; P 0.001), respectively. Among women, compared with patients without constipation, there was a 2.13-fold higher risk of AR in patients with constipation (aHR; 95% CI, 2.05-2.20; P 0.001). Among men, there was 2.02-fold higher risk of AR in patients with constipation (aHR; 95% CI, 1.92-2.13; P 0.001). Furthermore, patients with constipation had a significantly higher likelihood of AR, regardless of sex, and with or without comorbidities including hyperlipidemia, hypertension, chronic kidney disease, chronic liver disease, diabetes, COPD, RA, dyspepsia, IBS, and anxiety. …………………………………………………………………………………

From the point of view of the levels of evidence in evidence-based medicine, to date, there have been no meta-analyses, randomized control trials, or cohort studies related to constipation and AR risk. There are only a few small-scale observational studies to support these associations in the literature. Considering that supportive data from previous studies rarely link constipation to AR, our study aimed to investigate the association between constipation and AR by a large nationwide population-based cohort design. We tried to overcome the limitations of previous studies by using a large sample size, adjusting for various comorbidities and a variety of medications, matching propensity scores to reduce potential confounding factors, which made it feasible to test our hypothesis. Such research is of importance to provide evidence for further basic and clinical research. We also included a shred of studies have suggested that atopy might be linked with constipation, and cited more evidences to support and reinforce the hypothesis of current research.

We revised the paragraph in introduction and in discussion section and added the sentences as below: 

In 2nd paragraph of introduction section: 

A shred of studies have suggested that atopic disease might be linked to constipation [6-8]. Studies have reported an indirect indication of a concurrence of constipation and atopy by demonstrating a high prevalence of coexistent allergic manifestations in constipated children investigated for cow milk allergy [9-10]. An epidemiologic survey by Tokunaga et al. showed that constipation was a relevant factor for AR development (adjusted odds ratio of 1.17) among 21802 high school students [11]. Jones et al. stated that the overlap of atopy in functional gastrointestinal disorders patients, and the risk of rhinitis in 342 constipated patients was 1.66 times higher than controls [8]. …………………………………………………………………………….

………………………………………………………………………………………….

In 3rd paragraph of discussion section: 

………………………………………………………………………………………….…………………………………………………………………………………………. Although Parthasarathy et al. have offered that mucosal microbiota analysis could discriminate 25 constipated adults from healthy controls with 94% accuracy [47]. At variance, a recent study mentioned that no disease specific separation was observed by PCoA and by calculation of diversity indices in constipated children and healthy matched controls [19]. However, both groups could be discriminated with 82% accuracy by ridge regression. Although supportive data from previous studies rarely link constipation to AR, our findings provide support to the hypothesis that constipation and AR might share a similar underlying etiological pathways related to dysbosis. Nevertheless, the mechanism responsible for constipation-mediated dysbiosis or immune dysfunction, which precipitates AR development, requires further in-depth study. 

I greatly appreciate your time and your kind comments again; your comments reinforced our manuscript and we did our best to respond to your comments carefully.

2. In Discussion, the authors described a lot about the results and speculations by other researchers’ studies. However, there are no discussion how these can support your data (i.e., the authors describe others’ speculations to explain your speculations). Please discuss about the presented data.

Response: 

Thank you for your suggestions. 

We had revised and discussed presented data in more detail. We added this paragraph for explanation as below in 2nd paragraph of discussion section: 

………………………………………………………………………………………….…………………………………………………………………………………………

Consistent with our finding that there was a significant correlation between constipation and allergic diseases, Palmieri, M. et al. found a significant difference in the prevalence of atopic diseases proven by skin prick tests between the constipated children and the control group (17/52 = 33% versus 11/74=15%; p = 0.03) [33]. In addition, Jones et al. demonstrated that the overlap of atopy among 23471 functional gastrointestinal disorders patients, and the risk of rhinitis was 1.66-fold higher in patients with constipation than controls [8]. We also noted that hyperlipidemia, hypertension, chronic kidney disease, chronic liver disease, diabetes, COPD, RA, dyspepsia, IBS, and anxiety were associated with greater risk of AR in patients with constipation. RA was a comorbidity with a relatively higher AR risk (HR: 2.68, 95% CI, 1.41-5.1) compared with other comorbidities. Both RA and AR are characterized by the regulatory T-cells dysfunction [34]. Besides, constipation may worsen pre-existing dysbiosis in patients with RA [35, 36]. Moreover, patients with autoimmunity appeared to be predisposed to subsequent AR in our study. However, the risk for AR of other autoimmune diseases such as SLE, AS, Sjogren’s syndrome did not reach statistical significance in subgroup analysis. There might have been insufficient statistical power to detect significant differences due to the low incidence of these autoimmune diseases in our patient population.

It has been mentioned in other studies that dysbiotic microbiome could affect both constipation and AR, therefore we speculated the possibility that this could be a mechanism behind the association. We have included in the discussion that it is a limitation that we do not have fecal microbiota assessment, but more research can be done to assess this possibility. 

3. Also, the background of study participants varies. For example, the authors included autoimmune diseases. Please describe more details (e.g., types of diseases), and explain or speculate how each autoimmune disease can be associated with AR and/or constipation (i.e., pathophysiological mechanisms).

Response: 

Thank you for your comments. We clarify the types of autoimmune diseases including systemic lupus erythematosus (SLE), rheumatoid arthritis (RA), Sjogren’s syndrome, ankylosing spondylitis (AS) and found that patients with autoimmunity appeared to be predisposed to subsequent AR in our study (as shown in table 2). RA was a comorbidity with a relatively higher AR risk (HR: 2.68, 95% CI, 1.41-5.1) compared with other comorbidities. We also explain the possible pathophysiological mechanisms as below in discussion: 

……………………………………………………………………………………………………………………………………………………………………………………RA was a comorbidity with a relatively higher AR risk (HR: 2.68, 95% CI, 1.41-5.1) compared with other comorbidities. Both RA and AR are characterized by the regulatory T-cells dysfunction [34]. Besides, constipation may worsen pre-existing dysbiosis in patients with RA [35, 36]. Moreover, patients with autoimmunity appeared to be predisposed to subsequent AR in our study. However, the risk for AR of other autoimmune diseases such as SLE, AS, Sjogren’s syndrome did not reach statistical significance in subgroup analysis. There might have been insufficient statistical power to detect significant differences due to the low incidence of these autoimmune diseases in our patient population.

We further cited reference as below:

34. Cooles FA, Isaacs JD, Anderson AE. Treg cells in rheumatoid arthritis: an update. Curr Rheumatol Rep. 2013;15(9):352.

35. Häger J, Bang H, Hagen M, Frech M, Träger P, Sokolova MV, et al. The Role of Dietary Fiber in Rheumatoid Arthritis Patients: A Feasibility Study. Nutrients. 2019;11(10). 

36. Horta-Baas G, Romero-Figueroa MDS, Montiel-Jarquín AJ, Pizano-Zárate ML, García-Mena J, Ramírez-Durán N. Intestinal Dysbiosis and Rheumatoid Arthritis: A Link between Gut Microbiota and the Pathogenesis of Rheumatoid Arthritis. J Immunol Res. 2017;2017:4835189.

4. I would request the authors to present more detail data showing direct, at least suggestive, evidences for your hypothesis (i.e., underlying mechanisms linking constipation and AR). 

Response: 

We further analyzed a variety of associated comorbidities and various medications (based on Nat Rev Dis Primers. 2017 Dec 14;3:17095) and matched propensity scores to reduce the bias of possible residual confounders. We found a significantly higher risk of AR (aHR, 2.30; 95% CI, 2.23-2.37) was observed in constipated cohort than non-constipated cohort and constipated patients had a significantly higher likelihood of AR, regardless of age, sex, and with or without comorbidities including hyperlipidemia, hypertension, chronic kidney disease, chronic liver disease, diabetes, COPD, RA, dyspepsia, IBS, and anxiety. We have presented more detail data in the revised manuscript. The association between constipation and AR have been suggested in a few studies. [8, 11, 33] We further cited more-recent studies to offer the evidences regarding constipation and dysbiosis in adults and children. In addition, some up-to-date research of dysbiotic fecal microbiota in AR patients has been added to support the hypothesis that constipation and AR might share potential etiological pathways related to dysbiosis. We revised the paragraph of discussion regarding possible pathophysiological mechanisms as below. However, the observational design of current research precluded to elucidate the mechanisms underlying these associations, this is an inherent limitation of population-based datasets such as the NHIRD, future studies are warranted to validate our findings. (such as state-of-the-art metagenomic and metabolomic analyses of the gut microbiota in constipated patients are warranted to elucidate the possible pathogenetic mechanisms underlying these associations.) 

In 2nd paragraph of discussion section: 

…………………………………………………………………………………………………………………………………………………………………………………….. Consistent with our finding that there was a significant correlation between constipation and allergic diseases, Palmieri, M. et al. found a significant difference in the prevalence of atopic diseases proven by skin prick tests between the constipated children and the control group (17/52 = 33% versus 11/74=15%; p = 0.03) [33]. In addition, Jones et al. demonstrated that the overlap of atopy among 23471 functional gastrointestinal disorders patients, and the risk of rhinitis was 1.66-fold higher in patients with constipation than controls [8]. We also noted that hyperlipidemia, hypertension, chronic kidney disease, chronic liver disease, diabetes, COPD, RA, dyspepsia, IBS, and anxiety were associated with greater risk of AR in patients with constipation……………………………………………………………………………..

In 3rd paragraph of discussion section: 

 The pathophysiology underlying the relationship between constipation and subsequent AR remain ………………………………………………………................

.………………………………………………………………………………………….…………………………………………………………………………………………..Although Parthasarathy et al. have offered that mucosal microbiota analysis could discriminate 25 constipated adults from healthy contols with 94% accuracy [47]. At variance, a recent study mentioned that no disease specific separation was observed by PCoA and by calculation of diversity indices in constipated children and healthy matched controls [19]. However, both groups could be discriminated with 82% accuracy by ridge regression. Although supportive data from previous studies rarely link constipation to AR, our findings provide support to the hypothesis that constipation and AR might share a similar underlying etiological pathways related to dysbiosis. Nevertheless, the mechanism responsible for constipation-mediated dysbiosis or immune dysfunction, which precipitates AR development, requires further in-depth study. 

8. Jones MP, Walker MM, Ford AC, Talley NJ. The overlap of atopy and functional gastrointestinal disorders among 23,471 patients in primary care. Aliment Pharmacol Ther. 2014;40: 382-391.

11. Tokunaga T, Ninomiya T, Osawa Y, Imoto Y, Ito Y, Takabayashi T, et al. Factors associated with the development and remission of allergic diseases in an epidemiological survey of high school students in Japan. Am J Rhinol Allergy. 2015;29(2): 94-99.

33. Palmieri M, Ardia E, Caro MA, Palmieri S. Constipation and atopic diseases in children. Italian Journal of Allergy and Clinical Immunology. 2009;19:39-43.

Besides, we further cited reference as below: 

19. de Meij TG, de Groot EF, Eck A, Budding AE, Frank Kneepkens CM, Benninga MA, et al. Characterization of microbiota in children with chronic functional constipation. PLoS One. 2016;11(10): e0164731.

24. Liu X, Tao J, Li J, Cao X, Li Y, Gao X, et al. Dysbiosis of Fecal Microbiota in Allergic Rhinitis Patients. Am J Rhinol Allergy. 2020:1945892420920477.

27. Kim WG, Kang GD, Kim HI, Han MJ, Kim DH. Bifidobacterium longum IM55 and Lactobacillus plantarum IM76 alleviate allergic rhinitis in mice by restoring Th2/Treg imbalance and gut microbiota disturbance. Benef Microbes. 2019;10(1):55-67.

47. Parthasarathy G, Chen J, Chen X, Chia N, O'Connor HM, Wolf PG, et al. Relationship Between Microbiota of the Colonic Mucosa vs Feces and Symptoms, Colonic Transit, and Methane Production in Female Patients With Chronic Constipation. Gastroenterology. 2016;150(2):367-79.

We thank again from you for your best comments. Dear editor, your comment helps us to promote quality of this manuscript and we appreciate your kind suggestions.

---

## [Decision Letter · Decision Letter 1]

27 Aug 2020

PONE-D-20-09268R1

Constipation is associated with risk of allergic rhinitis: A nationwide population-based cohort study

PLOS ONE

Dear Dr. Wei,

Thank you for submitting your manuscript to PLOS ONE. After careful consideration, we feel that it has merit but does not fully meet PLOS ONE’s publication criteria as it currently stands. Therefore, we invite you to submit a revised version of the manuscript that addresses the points raised during the review process.

Your paper was reviewed by three experts in the field and myself. Although the quality of paper is improved, there are still some issues for analyses. Please answer for the comments by the reviewers (clarification of statistical issues). Also, the authors still overstate the results of statistical analyses (e.g., "Constipation was significantly associated with an increased risk of incidental AR"). It should be "might be associated." Since the authors did not show any clinical and experimental evidences to link these two conditions, please tone down and modify these statements seen everywhere in the paper including the title. The authors should strongly emphasize the limitation of study (e.g., lack of laboratory data including cytokines and gene-expression changes, etc.). 

We look forward to receiving your revised manuscript.

Kind regards,

Tomohiko Ai, M.D., Ph.D.

Academic Editor

PLOS ONE

Reviewers' comments:

Reviewer's Responses to Questions

**Comments to the Author**

1. If the authors have adequately addressed your comments raised in a previous round of review and you feel that this manuscript is now acceptable for publication, you may indicate that here to bypass the “Comments to the Author” section, enter your conflict of interest statement in the “Confidential to Editor” section, and submit your "Accept" recommendation.

Reviewer #1: All comments have been addressed

Reviewer #2: (No Response)

Reviewer #3: All comments have been addressed

2. Is the manuscript technically sound, and do the data support the conclusions?

Reviewer #1: Yes

Reviewer #2: Yes

Reviewer #3: Yes

3. Has the statistical analysis been performed appropriately and rigorously? 

Reviewer #1: Yes

Reviewer #2: Yes

Reviewer #3: Yes

4. Have the authors made all data underlying the findings in their manuscript fully available?

Reviewer #1: Yes

Reviewer #2: Yes

Reviewer #3: Yes

5. Is the manuscript presented in an intelligible fashion and written in standard English?

Reviewer #1: Yes

Reviewer #2: Yes

Reviewer #3: Yes

6. Review Comments to the Author

Reviewer #1: Dear Colleagues,

It is really interesting and well done work.This study after corrections is more interesting, sounds better and the data are presented clearer. However I have another comment (minor): I think the conclusion is slightly too restrictive , especially second sentence... Of course there were a great number of patients , good statistical tools but it was retrospective analysis and not meta analysis too. Therefore I propose be less restrictive in conclusion and use a term: it seems..."

Best regards

Reviewer #2: A propensity-matched analysis was conducted to show the association of constipation on time to allergic rhinitis (AR) development. After adjusting for relevant factors, patients with constipation had a 2.3 fold risk of AR compared to those without constipation.

Minor revisions:

1- Abstract: Methods: Replace the word “stratified” because the term causes confusion when it is not used based on its statistical definition. Consider using the term subgroup instead.

2- Line 153: Provide a reference for the specific software procedure used to conduct the propensity-score matching.

3- Line 159: For the Kaplan-Meier analysis indicate the start time.

4- In the statistical methods section, state the statistical testing methods used to generate the p-values in Table 1.

5- Line 216: Indicate the methods used to retain the adjustment variables of age, gender, comorbidities and medications in the model.

6- Table 2: For age, the multivariate Cox model should provide an overall p-value for comparing all the categories of age. If significant, a step-down test can be used to make pairwise comparisons. Indicate the overall p-value.

7- Table 3: (a) Provide a more descriptive title. See note #1 above.

(b) Table 3: Provide a description for each test of interaction by indicating the variables in the interaction term.

(c) Table 3: Clarify what the p-values represent. For instance, in those age 20 the log-rank p-value for comparing the time to allergic rhinitis between those with constipation and those without is 0.001. Is this interpretation accurate?

(d) In general, if the interaction effect is significant, provide an interpretation of the results, but do not test main effects because the tests for main effects are uninteresting in light of significant interactions. If interaction effects are non-significant, drop the interaction effects from the model and test the main effects. Determining which results to present when testing interactions is often a multi-step process.

8- Figure 2: Indicate the start time on the x-axis. “Years from xx.)

Reviewer #3: This is a large epidemiological study using the National Health Insurance Research Database in Taiwan. Propensity score matching is used to analyze the association between constipation and allergic rhinitis, taking into account possible confounding.

Despite the limitations of this study, I think that the results of the study suggest an association between gut microbiota and allergic rhinitis.

7. PLOS authors have the option to publish the peer review history of their article (what does this mean?). If published, this will include your full peer review and any attached files.

Reviewer #1: No

Reviewer #2: No

Reviewer #3: No

---

## [Author Response · Author response to Decision Letter 1]

9 Sep 2020

Tomohiko Ai, M.D., Ph.D.

Academic Editor

Journal: PLOS ONE 

Dear Dr. Tomohiko Ai 

On behalf of my coauthors, I thank the Editor/Reviewers for the comments and am grateful for the opportunity to revise our manuscript (ID: PONE-D-20-09268 R1) titled, “Constipation is associated with risk of allergic rhinitis: A nationwide population-based cohort study”, for consideration of publication as an Original Research in PLOS ONE. Your comments were highly insightful and enabled us to improve the quality of our document. In the following pages are our responses to each comment.

Revisions in the text are shown [yellow highlights]. We have addressed the comments point-by-point, and the corresponding changes have been made in track change in the manuscript, which you will find uploaded alongside this document. We hope that our revisions to the document combined with our accompanying responses will be sufficient to render our document suitable for publication in PLOS ONE.

Kind regards,

James Cheng-Chung Wei, MD, PhD

Institute of Medicine, Chung Shan Medical University 

No. 110, Sec 1, Jianguo N. Road, Taichung, 40201, Taiwan

jccwei@gmail.com

Responses to the Editor’s Comments:

1. Also, the authors still overstate the results of statistical analyses (e.g., "Constipation was significantly associated with an increased risk of incidental AR"). It should be "might be associated." 

Response:

Thank you for your comments and suggestions. We have toned down and modified these statements seen everywhere in the paper including the title. 

For example, we have revised the wording in conclusion of abstract: 

“Constipation might be associated with an increased risk of incidental AR.”

2. The authors should strongly emphasize the limitation of study (e.g., lack of laboratory data including cytokines and gene-expression changes, etc.). 

Response:

Thank you for your comments and suggestions. We have added the wording in limitations of discussion section:

“Thirdly, relevant clinical information, such as laboratory data including cytokines and gene-expression changes, imaging findings, and fecal microbiota assessments were unavailable in the database and therefore could not be included in the analyses.” 

We thank again from you for your best comments. Dear editor, your comments help us to promote quality of this manuscript and we appreciate your kind suggestions.

Responses to the Reviewers’ Comments:

Manuscript ID: PONE-D-20-09268 R1

Manuscript title: Constipation is associated with risk of allergic rhinitis: A nationwide population-based cohort study

Journal: PLOS ONE

Editor/Reviewer(s)' Comments to Author:

Reviewer #1: It is really interesting and well done work. This study after corrections is more interesting, sounds better and the data are presented clearer. However I have another comment (minor): I think the conclusion is slightly too restrictive, especially second sentence... Of course there were a great number of patients, good statistical tools but it was retrospective analysis and not meta analysis too. Therefore I propose be less restrictive in conclusion and use a term: it seems..."

Best regards

Response: 

Thank you for your comments and suggestions. 

We have changed the wording in conclusion as “It seems that physicians should keep a higher index of suspicion for AR in people with constipation.”

We thank you again for your best comments. Dear reviewer, your comments help us to promote quality of this manuscript and we appreciate your kind suggestions.

Reviewer #2: A propensity-matched analysis was conducted to show the association of constipation on time to allergic rhinitis (AR) development. After adjusting for relevant factors, patients with constipation had a 2.3 fold risk of AR compared to those without constipation. 

Minor revisions:

1- Abstract: Methods: Replace the word “stratified” because the term causes confusion when it is not used based on its statistical definition. Consider using the term subgroup instead.

Response:

Thank you for your comments and suggestions. We have changed the wording as suggested in line 38 of abstract by using the term “subgroup”

2- Line 153: Provide a reference for the specific software procedure used to conduct the propensity-score matching.

Response: 

Thank you for your comments and suggestions. We have attached the reference as suggested in line 154 and in the reference section. 

Reference:

Austin PC. An Introduction to Propensity Score Methods for Reducing the Effects of Confounding in Observational Studies. Multivariate Behav Res. 2011;46(3):399-424. 

3- Line 159: For the Kaplan-Meier analysis indicate the start time.

Response: 

Thank you for your comments and suggestions. We have added the wording as suggested in line 162 “….was used to calculate the cumulative incidence of AR from the index date”.

4- In the statistical methods section, state the statistical testing methods used to generate the p-values in Table 1. 

Response: 

Thank you for your comments and suggestions. We have added in the statistical methods section and p-values in Table 1.

5- Line 216: Indicate the methods used to retain the adjustment variables of age, gender, comorbidities and medications in the model. 

Response: Thank you for your comments and suggestions. 

 In general, from a statistical point of view, we could select the covariate p 0.2 into multivariate analysis through univariate analysis. However, this might ignore the potential confounding factors in this work. On the other hand, from the perspective of study design and related research using the NHIRD dataset, as well as the comorbid conditions and medications suggested by Editor, all the covariates of age, gender, comorbidities and medications were adjusted in multivariate analysis in this study. 

 In addition to the baseline comorbidities used in our research, other more important variables for constipation and developing AR as editor’s suggestions were worth exploring. We have analyzed and adjusted a variety of comorbid conditions and medications in detail based on the reference (Nat Rev Dis Primers. 2017 Dec 14;3:17095) suggested by Editor’s comments in R1 revision. Finally, we also considered the most chronic gastrointestinal conditions, such as dyspepsia, GERD, IBS, colonic polyps, GI tract cancers. Psychiatric/Mood disorders such as anxiety and depression; autoimmune diseases including SLE, RA, AS, Sjogren’s syndrome; neurological diseases such as Parkinson’s disease, multiple sclerosis, spinal cord injury and hypothyroidism were also enrolled for analysis. In addition, we further considered various medications associated with constipation including antihistamines, corticosteroids, non-steroidal anti-inflammatory drugs, calcium channel blockers, diuretics, opioids, antidepressants, serotonin antagonists, anticonvulsants, antispasmodic, iron supplement, calcium supplement. All covariates were adjusted in multivariate analysis.

We have added the wording below table 2:

Multiple Cox proportional hazard regression was used to adjust for age, gender, comorbidities, and medications. 

6- Table 2: For age, the multivariate Cox model should provide an overall p-value for comparing all the categories of age. If significant, a step-down test can be used to make pairwise comparisons. Indicate the overall p-value. 

Response: 

Thank you for your comments and suggestions. We have added the overall p-value of age in table 2. 

7- Table 3: 

(a) Provide a more descriptive title. See note #1 above.

Response: 

We have revised the table 3 title as “Subgroup analysis of hazard ratios (95% CI) of AR for patients with and without constipation by age, gender, and comorbidities.”

(b) Table 3: Provide a description for each test of interaction by indicating the variables in the interaction term.

Response:

Thank you for your comments and suggestions. The interaction effect was to compare the hazard ratios between different subgroups. To make the description more clear, we had added a footnote in table 3 as †p for difference of HR between subgroup and dropped the word “for interaction” in table 3.

And also added the description in results section: 

The interaction effect was to compare the hazard ratios between different subgroups. In subgroup of non-hypertension, non-hyperlipidemia, non-chronic liver disease, and non-depression, constipation group had significant higher risk of allergic rhinitis”.

(c) Table 3: Clarify what the p-values represent. For instance, in those age 20 the log-rank p-value for comparing the time to allergic rhinitis between those with constipation and those without is 0.001. Is this interpretation accurate?

Response: 

Thank you for your comments and suggestions. The p-value was from the Cox proportional hazard regression. This interpretation was constipation cohort had 2.09 fold risk of allergic rhinitis compared with non-constipation cohort in age 20. And the p value was 0.001.

(d) In general, if the interaction effect is significant, provide an interpretation of the results, but do not test main effects because the tests for main effects are uninteresting in light of significant interactions. If interaction effects are non-significant, drop the interaction effects from the model and test the main effects. Determining which

results to present when testing interactions is often a multi-step process.

Response: 

Thank you for your comments and suggestions. The interaction effect was to compare the hazard ratios between different subgroups. We had added a footnote in table 3 as † p for difference of HR between subgroup and dropped the word “for interaction” in table 3. 

8- Figure 2: Indicate the start time on the x-axis. “Years from xx.) 

Response: 

Thank you for your comments and suggestions. We have changed the x-axis as “Years from the index date”.

We thank you again for your best comments and suggestions. Dear reviewer, your comments help us to promote quality of this manuscript and we appreciate your kind suggestions.

Reviewer #3: This is a large epidemiological study using the National Health Insurance Research Database in Taiwan. Propensity score matching is used to analyze the association between constipation and allergic rhinitis, taking into account possible confounding. Despite the limitations of this study, I think that the results of the study suggest an association between gut microbiota and allergic rhinitis.

Response: 

Thank you for your comments and suggestions. 

Dear reviewer, I greatly appreciate your time and your kind comments again; your comments reinforced our manuscript and we did our best to respond to your comments carefully.

---

## [Decision Letter · Decision Letter 2]

14 Sep 2020

Constipation might be associated with risk of allergic rhinitis: A nationwide population-based cohort study

PONE-D-20-09268R2

Dear Dr. Wei,

We’re pleased to inform you that your manuscript has been judged scientifically suitable for publication and will be formally accepted for publication once it meets all outstanding technical requirements.

Kind regards,

Tomohiko Ai, M.D., Ph.D.

Academic Editor

PLOS ONE

Additional Editor Comments (optional):

Reviewers' comments:

Reviewer's Responses to Questions

**Comments to the Author**

1. If the authors have adequately addressed your comments raised in a previous round of review and you feel that this manuscript is now acceptable for publication, you may indicate that here to bypass the “Comments to the Author” section, enter your conflict of interest statement in the “Confidential to Editor” section, and submit your "Accept" recommendation.

Reviewer #1: All comments have been addressed

Reviewer #2: All comments have been addressed

2. Is the manuscript technically sound, and do the data support the conclusions?

Reviewer #1: Yes

Reviewer #2: (No Response)

3. Has the statistical analysis been performed appropriately and rigorously? 

Reviewer #1: Yes

Reviewer #2: (No Response)

4. Have the authors made all data underlying the findings in their manuscript fully available?

Reviewer #1: Yes

Reviewer #2: (No Response)

5. Is the manuscript presented in an intelligible fashion and written in standard English?

Reviewer #1: Yes

Reviewer #2: (No Response)

6. Review Comments to the Author

Reviewer #1: I think this form of ms is better. All comments were incorporated. The tables and statistical analysis and its description sounds also better.

Reviewer #2: (No Response)

7. PLOS authors have the option to publish the peer review history of their article (what does this mean?). If published, this will include your full peer review and any attached files.

Reviewer #1: No

Reviewer #2: No

---

## [Editor Report · Acceptance letter]

16 Sep 2020

PONE-D-20-09268R2 

Constipation might be associated with risk of allergic rhinitis: A nationwide population-based cohort study 

Dear Dr. Wei:

I'm pleased to inform you that your manuscript has been deemed suitable for publication in PLOS ONE. Congratulations! Your manuscript is now with our production department. 

Kind regards, 

on behalf of

Dr. Tomohiko Ai 

Academic Editor

PLOS ONE